# CD146^+^ Pericytes Subset Isolated from Human Micro-Fragmented Fat Tissue Display a Strong Interaction with Endothelial Cells: A Potential Cell Target for Therapeutic Angiogenesis

**DOI:** 10.3390/ijms23105806

**Published:** 2022-05-22

**Authors:** Ekta Manocha, Alessandra Consonni, Fulvio Baggi, Emilio Ciusani, Valentina Cocce, Francesca Paino, Carlo Tremolada, Arnaldo Caruso, Giulio Alessandri

**Affiliations:** 1Section of Microbiology, Department of Molecular and Translational Medicine, University of Brescia Medical School, 25123 Brescia, Italy; arnaldo.caruso@unibs.it (A.C.); cisiamo2@yahoo.com (G.A.); 2Neurology IV—Neuroimmunology and Neuromuscular Diseases Unit, Fondazione IRCCS Istituto Neurologico Carlo Besta, 20133 Milan, Italy; alessandra.consonni@istituto-besta.it (A.C.); fulvio.baggi@istituto-besta.it (F.B.); 3Laboratory of Neurological Biochemistry and Neuropharmacology, Fondazione IRCCS Istituto Neurologico Carlo Besta, 20133 Milan, Italy; emilio.ciusani@istituto-besta.it; 4CRC StaMeTec, Department of Biomedical, Surgical and Dental Sciences, University of Milan, 20122 Milan, Italy; valentina.cocce@unimi.it (V.C.); francesca.paino@unimi.it (F.P.); 5Department of Stem Cells and Regenerative Medicine, Image Institute, 20122 Milan, Italy; carlo.tremolada@gmail.com

**Keywords:** angiogenesis, pericytes, adipose tissue, cell adhesion, endothelial cells

## Abstract

Pericytes (PCs) are mesenchymal stromal cells (MSCs) that function as support cells and play a role in tissue regeneration and, in particular, vascular homeostasis. PCs promote endothelial cells (ECs) survival which is critical for vessel stabilization, maturation, and remodeling. In this study, PCs were isolated from human micro-fragmented adipose tissue (MFAT) obtained from fat lipoaspirate and were characterized as NG2^+^/PDGFRβ^+^/CD105^+^ cells. Here, we tested the fat-derived PCs for the dispensability of the CD146 marker with the aim of better understanding the role of these PC subpopulations on angiogenesis. Cells from both CD146-positive (CD146^+^) and negative (CD146^−^) populations were observed to interact with human umbilical vein ECs (HUVECs). In addition, fat-derived PCs were able to induce angiogenesis of ECs in spheroids assay; and conditioned medium (CM) from both PCs and fat tissue itself led to the proliferation of ECs, thereby marking their role in angiogenesis stimulation. However, we found that CD146^+^ cells were more responsive to PDGF-BB-stimulated migration, adhesion, and angiogenic interaction with ECs, possibly owing to their higher expression of NCAM/CD56 than the corresponding CD146^−^ subpopulation. We conclude that in fat tissue, CD146-expressing cells may represent a more mature pericyte subpopulation that may have higher efficacy in controlling and stimulating vascular regeneration and stabilization than their CD146-negative counterpart.

## 1. Introduction

Mesenchymal Stromal Cells (MSCs) are medicinal signaling cells involved in tissue regeneration but are not stem cells [1]. This cell population, which can be isolated from many tissues, such as bone marrow, placenta, adipose tissue (AT), and umbilical cord blood, has been shown to be identical to the vascular associated cell phenotype, namely pericytes (PCs) [2]. Indeed, both MSCs and PCs express similar markers and, most importantly, display very similar functional activity. They both are considered safe for allogenic transplantation due to the lack of expression of membrane-bound molecules involved in immune rejection [3]. MSCs have the potential to turn out as drug stores and be involved in immune modulation, tissue regeneration, and secretion of angiogenic molecules [4,5]. They prevent inflammatory cascades by secreting cytokines and paracrine factors to interact directly with different immune cells to suppress the over-activation of the immune system [6,7,8]. Overall, they appeared as a quite heterogeneous cell population that includes stromal cells at different levels of maturation and differentiation [7,8]. More specifically, PCs are vascular mural cells that interact with the abluminal surface of endothelial cells (ECs) from capillaries, arterioles, and venules and share a common basement membrane with ECs. They can be found around blood capillaries, precapillary arterioles, postcapillary venules, and collecting venules and are morphologically distinct in different organs [9]. Moreover, PCs have been found to regulate capillary diameter, blood flow, vessel permeability, and stabilization of ECs [10]. 

The synergism between the ECs and PCs in maintaining a functional vasculature is well studied in terms of regulation of different stages of vessel maturity as PCs may regulate both destabilization of nascent vessels and stabilization of mature vasculature [11,12]. Endothelial-free PC assemblies were observed to regulate sprouting in retinal neovascularization, adult mouse cornea, and mouse tumor models, by recruiting ECs from the parental vessel, signifying the guidance exerted by PCs for invading ECs [13]. Genetic or acquired deficiencies in pericyte coverage of EC-lined capillaries can lead to the instability of micro-vessels [14].

Little is known about the “maturity stage” of PCs involved in ECs migration or its role in mediating angiogenesis. The primary reason for this lack of knowledge is probably related to the fact that PCs are a heterogeneous cell population, difficult to characterize due to the absence of specific markers. Usually, PCs are characterized by many molecular markers [15], among them, NG2, PDGF-β receptor, and Endoglin (CD105) are indicated as the most common ones. In addition, CD146 which was originally identified as an endothelial marker involved in the angiogenesis process was also found significantly expressed in PCs [2,16]. Interestingly, it has been known that CD146^+^ stem cells appear to have a greater therapeutic potential than CD146^−^ stem cells in inflammatory diseases [17], as well as CD146 expression on MSCs was associated with their vascular smooth muscle commitment [18]. However, the role of these two PC populations on angiogenesis is poorly investigated.

Therefore, in this study, we investigated PCs distinguished by the presence of CD146 in regulating interaction with ECs and their efficacy on the angiogenesis process in vitro. PCs were isolated from AT upon a process of mechanical micro-fragmentation which allows obtaining a significant number of mesenchymal cells/pericytes [19,20]. Briefly, we found that CD146^+^ PCs were highly effective in interacting with ECs compared to their CD146-negative counterpart and consequently were more efficient in stimulating angiogenesis. We also postulated that the variable expression of CD56/NCAM by CD146^+^ PCs was related to their capacity to adhere to endothelium. 

## 2. Results

### 2.1. Characterization of Pericytes (PCs)/Mesenchymal Stromal Cells (MSCs) from Adipose Tissue (AT)

The common pericyte-specific markers that correspond to MSCs are often used for the unambiguous identification of the cells from the stromal vascular fraction (SVF) of AT along with a meticulous analysis of morphological criteria. Magnetic-activated cell sorting (MACS)-based CD31 selection was performed on the stromal cell population extracted from collagenase-treated micro-fragmented adipose tissue (MFAT). The CD31^–^ subset cells (that were around 0.62 ± 0.08 × 10^6^/mL of MFAT) were investigated for typical MSC markers. Immunostaining results identified the extracted CD31-negative cells to be mostly NG2^+^, PDGFRβ^+^, CD105^+^, and slightly positive for αSMA (Figure 1A). Furthermore, the CD31^−^ cells were investigated for other common markers using flow cytometric analysis and were observed to be CD34^−^/CD105^+^/CD73^+^/CD90^+^/CD44^+^; thus, expressing the typical markers of MSCs derived from adipose tissue (Figure 1B) thereby demonstrating their PC origin. The same cells were also majorly positive for adhesion molecules such as intracellular adhesion molecule 1 (ICAM1) (Figure 1B) and Vascular Endothelial Cadherin (VECAD) (Appendix A). As flow cytometric analysis showed the majority of the CD31^−^ cells as negative for CD146 (Appendix A), to separate this cell population we performed a second cell sorting (MACS)-based CD146 selection. The isolated CD146^+^ cells showed no morphological difference from the CD146^−^ cell populations. Nonetheless, it is important to have both CD146 subpopulations phenotypically different from CD31^+^ cells (Figure 1C) since the CD146 marker has also been shown to be expressed by ECs [21]. Consequently, both CD146^−^ and CD146^+^ subpopulations were screened for typical MSC markers by flow cytometry (Appendix A). CD146^+^ cells showed higher expression levels of ICAM1, CD44, CD105, and CD90 while CD146^–^ cells expressed a slightly higher proportion of CD73 surface marker (Figure 2A). In this regard, CD31 initial selection was crucial for removing all the ECs present in the MFAT tissue preparations; thus, excluding the possible presence of residual ECs contaminating the isolated PCs population. This was also confirmed by the negative expression of von Willebrand factor (data not shown), while both of the CD146-sorted populations do express characteristic MSC markers.

Under our experimental conditions, we found that the number of the isolated CD31^–^CD146^−^ cells was around 0.51 ± 0.09 × 10^6^ (per 5 mL of MFAT) and was almost 4–5-fold that of CD31^–^CD146^+^ cells (0.13 ± 0.02 × 10^6^ per 5 mL of MFAT). 

### 2.2. Fat Pericytes Respond to Platelet Derived Growth Factor (PDGF-BB) Signaling

Pericytes have been identified as important regulators of ECs signaling and vascular patterning as the release of platelet derived growth factor (PDGF-BB) by ECs activates PDGFR on PCs which, in turn, controls angiogenesis by regulating vascular endothelial growth factor (VEGF-A)/VEGFR-2 signaling [12,22]. Fibroblast growth factor (FGF-2) binds to FGFR2 to stimulate pericyte proliferation and orchestrates the PDGFRβ signaling, directly and indirectly, for vascular recruitment. In angiogenic vessels, ECs produce PDGF-BB to recruit PDGFRβ^+^ pericytes onto the nascent vasculature [23,24].

To this aim, we asked whether CD31^–^ PCs can sustain differential migratory potential toward common growth factors. Cultured CD31^–^PCs from fat tissue exhibited strong chemotaxis towards FGF-2 stimulation (Figure 2B). However, they were not found to be stimulated with VEGF-A signals, therefore, speculating towards basic FGF mediated downstream signaling in a collagen-containing medium. The migration response to FGF-2 was comparable to the positive control, stimulated with endothelial growth medium (EGM) in the presence of serum and other growth factors. The negative control, on the other hand, basal medium without the presence of any growth factors, appeared to be quite slow to respond to the migratory stimuli. 

PDGF is released from angiogenic ECs and the binding of PDGF-BB to PDGFRβ on the pericytes induces their migration by activating the Ras/Rho/Rac and protein kinase C pathway [9]. Since FGF-2 activates downstream PDGF signaling and owing to the same origin of CD146-sorted PCs, we attempted to test the different migratory properties of CD146^+^ and CD146^−^ subpopulation upon stimulation with PDGF-BB. CD146^+^ PCs were found to migrate and proliferate much higher in number than CD146^−^ cells in response to PDGF-BB in the basal medium (Figure 2C). To our surprise, CD146^+^ PCs migrated much slower than their CD146^−^ counterparts upon collagen embedding in a complete medium in the absence of PDGF-BB which was evident starting from day 4 until day 7 post-stimulation (Appendix A). 

### 2.3. Pericytes from Fat Tissue Interact with Human Umbilical Vein ECs (HUVECs)

The interaction of PCs with ECs is crucial in maintaining the mechanical stability of micro-vessels and can be studied with the help of adhesion assays. For this reason, we determined the adhesion of CD31^−^ PCs to the surface of quiescent ECs monolayer at 15, 30, and 60 min of incubation where 30 min was chosen as the optimum time point for the identification of round-shaped PCs attached to the resting elliptical-shaped ECs (Appendix A). At the mentioned time point, a reasonable number of PCs among both subpopulations were observed to be in the vicinity of ECs. As a negative control, only HUVECs, without the presence of PCs, were used (Figure 3A). Interestingly, the adhesion of CD146^−^ PCs to the HUVECs monolayer was significantly lower than that of CD146^+^ PCs (Figure 3A) suggesting that CD146^+^ PCs may have much more efficacy than their CD146^−^ counterpart in the angiogenesis process, particularly in remodeling and stabilization of ECs during micro-vessels neoformation.

### 2.4. Micro Fragmented Adipose Tissue (MFAT)-Derived CD146^+^ PCs Subset Promotes Vascular Stability

We initially investigated if the conditioned medium (CM), obtained by incubation of the whole PCs population derived from MFAT tissue and cultured at very low serum concentration in basal medium, was able to preserve ECs vascular monolayer integrity. Thus PCs-CM was added (2-fold diluted) to the HUVECs monolayer and then cultured under starvation for 24 h. PCs-CM demonstrated potent efficacy in preserving ECs monolayer integrity. The effect of CM was equal to that of the complete growth medium (EGM + 10% FBS). In contrast, the EC monolayer cultured under starved conditions (medium consisting of only EBM + 0.5% FBS) and in the absence of PCs-CM was significantly damaged: cells were non-viable and almost completely detached as observed by the presence of floating cells in the culture (Figure 3B). Furthermore, ECs stimulated with PCs-CM up to a dilution of 64-fold were able to survive under starved conditions (Appendix A).

Once the efficacy of PCs on vascular monolayer integrity was established, we next performed a 3-D spheroids assay by mixing CD31^−^ PCs and ECs in a ratio of 1:5. As shown in Figure 4A, the presence of PCs not only perturbated the angiogenesis process but also aggravated the same as observed by a higher number of sprouts formed by HUVECs. This strong effect was also confirmed by CD31^+^ endothelial cells derived from the fat tissue itself. Then, we asked if the two subpopulations of PCs, CD146^−^ and CD146^+^, could modulate vascular sprouting and stabilization differently. We found that the induction of sprouting by ECs combined with the CD146^+^ cell population was, in general, higher than that induced by CD146^−^, but not statistically different (Figure 4B). However, CD146^+^ cells induced a significantly higher sprout length than CD146^−^ cells (Figure 4C) suggesting that CD146^+^ cells may have been more tightly integrated with ECs as observed by longer cytoplasmic extensions to the spheroid core and therefore higher capacity to support ECs stability. This was confirmed by an investigation of capillary-like structures via tube formation assay. In this case, we tested the effect of CM derived from cultured CD146^+^ and CD146^−^ fat PCs on HUVECs seeded on polymerized plugs of growth factor reduced (GFR)-basement membrane extract (BME). The addition of CD146^+^ PCs-derived CM was more effective as compared to CD146^−^-CM to induce the significant production of cord-like formation by ECs when seeded on the Matrigel. Indeed, the effect of CD146^+^-CM was most effective after 24 h of incubation. At this time of observation, while cord formation persisted in CD146^+^-CM-treated wells, the cords were completely regressed in the control medium and CD146^−^-CM (1:2 dilution) treated wells (Figure 4D). Interestingly, under these experimental conditions, the addition of CM derived from CD146^+^ PCs was much more effective than the CM of the CD146^−^ PC subset in stimulating EC sprouting number in a 3-D spheroids assay (Figure 4E). 

### 2.5. Upregulation of CD56 (NCAM) Expression by CD146^+^ PCs

Pericyte–endothelial cell interaction is crucial in tissue regeneration which is related to the activation of signals that regulate endothelial cell function. Among the presence of other adhesion molecules such as ICAM1, VECAD, and TGFβ signaling as previously reported, we attempted to investigate if NCAM (CD56), another marker known to mediate endothelial cell–pericyte interactions [25] is also differentially expressed among CD146-sorted pericyte population. As shown in Figure 5A, CD146^+^ PCs relatively expressed NCAM as punctate staining in the cytoplasm. On the contrary, CD146^–^ PCs do not show a marked expression of NCAM. However, both the populations did exhibit PDGFRβ expression but CD146^+^ cells showed a relatively higher expression of the same in the cytoplasm and plasma membrane than their counterpart. Labeling studies demonstrated that CD146^+^ cells were strongly positive for CD44, CD56, and negative for αSMA, whereas the CD146^−^ cells stained strongly for CD44 but significantly less intense for CD56, and negative for αSMA (Figure 5B). Under our experimental conditions, we identified the higher expression of CD105 by CD146^+^ cells (also confirmed by flow cytometry) while a dim expression by CD146^−^ cells was observed whereas the two subpopulations that stained positive for NG-2 with CD146^+^ cells displayed a relatively stronger expression in both nucleus and cytoplasm (Figure 5C). Since a higher level of PDGFRβ receptor upregulates cell adhesion molecules in vitro and in vivo, we examined the NCAM expression of PCs upon adhesion to ECs. We found that CD146^+^ PCs, which adhere to a greater extent than CD146^−^ PCs, also appeared to express NCAM at a higher level (Figure 6B,C). The negative control IgG and IgG1 isotype stained negatively for anti-PDGFRβ and anti-NCAM antibodies, respectively (Figure 6A). Therefore, these experiments seem to confirm that NCAM may play a role in the interaction of PCs with endothelial cells and the CD146^+^ PCs subset may have an important role in mediating vascular stability.

## 3. Discussion

PCs are involved in various stages of angiogenesis including EC migration, proliferation, and subsequent endothelial tubulogenesis and vessel stabilization [26]. This implies that the interaction of ECs–PCs may influence the parenchymal–stromal cell cross-talk and may provide insights into specific treatment therapy [27]. In this study, we investigated two distinct subpopulations of PCs, isolated from human MFAT which represent a very rich tissue source for stromal cells [28,29], and, based on CD146 segregation, we analyzed their angiogenic activity along with their interaction with human vascular endothelial cells.

Owing to the heterogeneity of pericytes, a number of molecular markers have been used to identify this cell population. Crisan et al. validated that CD146^+^ NG2^+^ PDGFRβ^+^ ALP^+^ CD34^−^ CD45^−^ vWF^−^ CD144^−^ phenotype is an indicator of pericytes or perivascular cells throughout human fetal and adult organs mostly associated with capillaries and micro-vessels. Particularly, the presence of NG2 and CD146 mark myogenic progenitors at the periphery of larger veins and arteries or blood vessels [2]. We here identified MFAT-derived pericytes by the expression of CD146^+^, NG2^+^, PDGFRβ^+^, CD105^+^, CD73^+^, CD90^+^, CD44^+^, CD34, and mostly αSMA^−^. Alongside, we found another phenotypically similar population which is devoid of CD146. Comparing the two PC populations, we found that CD146-positive (CD146^+^) PCs were more adherent to the ECs monolayer than CD146-negative (CD146^−^) PCs and also expressed higher levels of molecules such as ICAM1 and CD105. This leads to the speculation that the distinction between the two populations of PCs from human AT may aid in elucidating the critical molecules involved in the adhesion or eventually vascularization process of ECs. Thus, supporting the idea that clarifying the mechanisms by which PCs may adhere to endothelial cells’ surfaces may better define their role in vascular remodeling and formation. However, the expression of MSC markers among the two subpopulations may vary in culture depending on the passage number, culture conditions, and differentiation potential.

PCs release VEGF that binds VEGFR2 on ECs, thereby recruiting them to mediate angiogenesis whereas Ang-1 and PDGF-BB regulate PC coverage of ECs differently in normal and pathological conditions [13,30]. Additionally, placental PCs are known to secrete significant quantities of HGF (hepatocyte growth factor) that can be a potent paracrine angiogenic stimulus for EC sprouting where PCs were recruited to the sprouts by PDGF-BB [14,31,32]. Interestingly, we here observed that CD146^+^ PCs respond very consistently and rigorously to PDGF-BB-stimulated migration in a concerted way, whereas they do not respond at the same pace to migration in the presence of multiple growth factors containing complete media. On the contrary, CD146^−^ PCs responded highly to different growth factors containing complete media but not enough to PDGF-BB stimulation. This observation corresponds to a low expression of PDGFβ-receptors on the surface of CD146-negative cells and possibly the variable expression of other growth factor receptors similar to microvascular cells. Furthermore, FGF-2 synchronizes with the PDGF-BB–PDGFRβ signaling pathway by modulating their expression and activation [24]. This was postulated in our results as MFAT-derived pericytes, in particular, respond positively to both FGF-2- and PDGF-BB-mediated migration but not the stimuli meant for ECs. Since CD146-expressing cells are known to lose their expression on expansion in vitro [21], we observed the same effect in our studies at higher passage (data not shown) while the differential migratory activity between both the subpopulations remained intact.

PCs have been long known to regulate angiogenesis either by direct contact or paracrine effect. In a previous study, the Diptheria toxin-mediated ablation of PCs led to morphological changes in endothelial sprouts at the leading edge of the vascular plexus with thicker and blunt-ended sprouts compared to slender morphology of sprouts in the presence of PCs [12]. Our study reports the difference between both populations of pericyte-driven angiogenesis where the mix of EC-CD146^−^ PCs is not able to carry out integrated sprouting of ECs, mimicking the pericyte ablation condition. Opposite to the former, EC-CD146^+^ PCs direct a sustained and robust sprouting pattern, thereby making CD146 an indispensable marker for pericyte-driven vessel stabilization. Of note, the destabilizing effect of low pericyte coverage can lead to inappropriate extensive angiogenesis [33] which may be the case for CD146^−^ PCs. 

In another finding, human muscle PCs were shown to inhibit cord formation by dermal microvascular cells through CXCR3-induced ECs involution [10]. On the other hand, CD146^+^ PCs have been reported to induce remodeling of vessels under circumstances such as tumor growth and invasion of hypoxia-induced angiogenesis [13,34,35]. Comparably, we identified MFAT-derived PCs to aggravate sprout formation by macrovascular endothelial cells. MFAT- and MSCs-secretome has already been identified to release factors such as β-FGF, HGF, IL-8, IL-16, and VEGF, SCGF-β, IL-6, and MCP-1, respectively [19]. Similarly, CM derived from CD146^+^ PCs facilitated higher tube formation as well as sprouting by endothelial cells than CD146^−^ PCs-CM, emphasizing that CD146^+^ PCs provide paracrine survival support for ECs.

Finally, NCAM (CD56) expression modulation has been implicated in the progression of different human cancers. TGF-β1 was shown to reduce the interaction between stromal cells and liver ECs through its capacity to down-modulate NCAM expression, thereby attesting to the important role of NCAM in pericyte–EC interaction and thus in vascular stability [25]. Since CD146^+^ PCs were observed to upregulate NCAM expression but not CD146^−^ cells, this information provides us with a rationale to support the importance of the stromal cells–EC interaction in mediating angiogenesis. To our knowledge, this is the first report which highlights the fat PCs as composed of CD146^−^/CD56^−^ and CD146^+^/CD56^+^ subpopulations in displaying a differential angiogenic activity. NCAM expression was evident in both resting and adhered states of CD146^+^ PCs, confirming the postulation that NCAM expression has a role in the interaction and/or adhesion of PCs with endothelial cells [36]. Since CD146 seems important for PDGFRβ-induced PCs recruitment [37], whether the expression of NCAM also has a potential role in driving PCs to upregulate the expression of PDGFRβ remains to be elucidated. Understanding the mechanism by which the pericyte population interacts with the endothelial cells to induce vascular stability represents a fundamental stem for the development of both pro- and anti-angiogenic therapies [24,38]. This study highlights the importance of the PCs subpopulation expressing CD146^+^/NCAM markers to represent a step forward in this direction.

## 4. Materials and Methods

### 4.1. Lipoaspiration

Samples of human MFAT were obtained by liposuction of subcutaneous tissue as previously described elsewhere by using disposable cannulas provided with the Lipogems^®^ kit [39,40]. Tissue samples were collected from plastic surgery operations after signed informed consent by the patient, in accordance with the Declaration of Helsinki. Written informed consent, specifying that residual material destined to be disposed of could be used for research, was signed by each participant before the biological materials were removed, in agreement with Rec (2006)4 of the Committee of Ministers Council of Europe on research on biological materials of human origin. The approval for their use was obtained from the Institutional Ethical Committee of Milan University (n.59/15, C.E. UNIMI, 09.1115).

For all the in vitro experiments performed in this study, the fat tissue was obtained from five different human donors (4 females and 1 male, age median 54 ± 7) that underwent plastic surgery. The fat tissue was harvested from the abdominal site. Each experiment was performed with the material obtained from a single donor and similar results were acquired with the material from other donors. The cell confluence rate was variable for different donors.

### 4.2. Cells and Media

PCs were maintained in EGM-2 MV media (Lonza, Basel, Switzerland) containing 10% FBS and growth factors (EGM-MV2 Bullet Kit, Lonza). PCs were passaged every 2 days. Cells were cultured until 70–80% confluence for all the experiments until passages 3-4. HUVECs were maintained in complete EGM MV media (Promo cell, Heidelberg, Germany) containing growth factors with 10% FBS. HUVECs were cultured until 70-80% confluence for all the experiments until passage 6. Wherever mentioned, ECs and PCs were starved in EBM (Promo cell) or EBM-2 (Lonza) basal medium containing 0.5% FBS.

### 4.3. Isolation and Cell Cultures of Pericytes from Fat Tissue

To discriminate the AT-derived MSCs (CD31^−^) from ECs (CD31^+^), 3–5 mL of fat samples (fresh MFAT specimens) were used. The MFAT was collected in 15 mL conical tubes and washed twice with RPMI-1640 media (Sigma, St. Louis, MO, USA) containing 0.2% BSA. The fat specimens were digested with collagenase (0.25% *w*/*v*, Sigma) to evaluate the total cells and MSC content. After collagenase digestion for 1 h at 37 °C, DMEM/F-12 medium (Gibco, Life Technologies, Monza, Italy) + 10% FBS was added to the tube to stop the enzymatic reaction followed by centrifugation for 5 min, 1200 rpm. The obtained cell pellets were processed for selection with CD31-magnetic microbeads (BD biosciences, Italy) as previously described [19] followed by culturing of cells in EGM-2 + 10% FBS for 5 days or until confluence. Once the cells reached confluence, they were further processed for selection with CD146-magnetic beads (Miltenyi Biotec, Germany) to distinguish CD146^+^ from the CD146^−^ population. Both CD146^+^ and CD146^−^ populations were maintained in culture and passaged every 2–3 days.

### 4.4. Immunophenotyping by Flow Cytometry

Phenotypical characterization of donor PCs was performed by multicolor flow cytometry. PCs were harvested after CD31-based selection followed by CD146 selection (as mentioned above), washed with EGM-2 complete medium followed by PBS, and incubated with PE- or FITC- or APC-conjugated antibodies for 1 h at 4 °C according to the manufacturer’s recommendation and then analyzed. CD31^−^ PCs were characterized with the following antibodies: anti-CD34-FITC, anti-CD44-APC, anti-CD73-APC, anti-CD90-FITC, anti-CD146-FITC (all from BD Pharmingen Franklin Lakes, NJ, USA); anti-CD105-PE (Immuno tools, Friesoythe, Germany); anti-ICAM1-PE and anti-VECAD-FITC (both from Biolegend, San Diego, CA, USA).

For CD146-selected PCs, CD34-FITC/CD105-PE/CD73-APC and CD146-FITC/ICAM1-PE/CD44-APC were detected as triple stains, while CD90-FITC was detected as a single stain. Isotype-matched nonreactive fluorochrome-conjugated antibodies were used as controls and quantitative analysis was performed using a Navios EX flow cytometer (Beckman Coulter, Brea, CA, USA) with software Navios (Beckman Coulter, Brea, CA, USA). In this study, at least 10,000 events were analyzed for each sample excluding non-viable cells based on forward scatter and side scatter parameters. The data are expressed as the ratio of mean fluorescence intensity (MFI) of each specific antibody and the relative isotype control. Values greater than 1 indicate the expression of the specific marker.

### 4.5. Immunohistochemical Staining for Pericyte Markers

In total, 50,000 cells were plated on 8-well chamber slides (Labtek, Thermofisher Scientific, Waltham, MA, USA) and fixed with 4% paraformaldehyde. Immunohistochemical analysis was performed as mentioned previously [24]. The antibodies used for this assay were anti-CD105 (1:50, Histo-line Laboratories, Italy), anti-NG2 (1:50, Santa Cruz Biotechnology, Dallas, TX, USA), anti-CD44 (1:50, Agilent Technologies, Santa Clara, CA, USA), anti-PDGFRβ (1:200, Santa Cruz Biotechnology), and anti-αSMA (1:1000, Biocare Medical, Italy). Digital photographs were obtained using the Olympus DP73 digital camera (Olympus Corporation, Milan, Italy).

### 4.6. Spheroids Assay

Spheroids were generated by mixing 0.2 × 10^5^ PCs and 1 × 10^5^ HUVECs in a 1:5 ratio in 10 mL of complete EGM with methylcellulose (Sigma) and incubated at 37 °C in 96-well (100 μL/well) non-adherent plates (Greiner Bio-one, Kremsmünster, Austria). The collagen solution was formed with Rat tail collagen I (Corning, NY, USA), 1X PBS, 1X NaOH, and final pH 7.4 after neutralization with 0.1 N NaOH. The next day, spheroids were collected, centrifuged at 2000× *g* for 10 s, resuspended in neutralized collagen solution, and plated on neutralized collagen-coated chamber slides. After 1 h of incubation, the cells were treated with complete EGM-2 medium or CM from PCs for overnight incubation at 37 °C. Wherever mentioned, CD31^+^ cells from fat tissue were used to generate spheroids. Sprouting was observed from the spheroid core, and the sprout length (mean  ±  SD) was measured using the Image-J software for at least five spheroids with similar sizes and sprout numbers from three wells/conditions.

### 4.7. Adhesion Assay

All cells were washed with RPMI media containing 0.2% BSA. In total, 6000 PCs were counted using a hemocytometer (Sigma), added in the same number for each condition to the HUVECs monolayer (30,000 cells) seeded in a 96-well plate, and incubated for 30 min at 37 °C, 5% CO_2_ in a humidified incubator. Afterward, cells were fixed and stained with a Diff-Quik staining kit (Medion Diagnostics, Düdingen, Switzerland) and the number of round-shaped PCs adhered to the elliptical-shaped HUVECs layer was analyzed and counted manually based on the phenotypic distinction in 3 different frames/well for at least 3 wells per condition. For the cytostaining, the HUVECs monolayer was seeded in 8-well chamber slides (Labtek, Thermofisher Scientific) for performing the adhesion assay. 

### 4.8. Collagen Migration Assay

A total of 25,000 PCs were detached from the flasks and resuspended in a mixture of collagen I, 1X DMEM/F-12 medium, and 1X sodium bicarbonate solution. Gel drops were then created in a 4-well plate (Nunc., Thermofisher Scientific) and incubated at 37 °C for 1 h. After incubation, cell-loaded collagen drops were bathed in a medium containing FGF-2 (50 ng/mL) (Santa Cruz Biotechnology) or VEGF-A (50 ng/mL) (Miltenyi Biotec). Collagen drops bathed in EBM-2 media with 0.5% serum or EGM-2 media with 10% FBS were used as a negative and positive control, respectively. PCs migration outside collagen drops was quantified as the distance covered at least after 72 h obtained with a DM-IRB microscope system (Leica, Wetzlar, Germany) and photographed with a Hitachi KP-D50 camera. The distance was measured manually as a parameter of length from the edge of the collagen drops to the leading edge of the cells.

### 4.9. Transwell Assay

Corning Costar Transwell supports were used to test spontaneous and PDGF-BB-stimulated pericytes migration. The 6.5 mm Transwell with 5 μm pore size polycarbonate membrane inserts was coated with Coll-1 as previously described [41]. For each test, 1 × 10^5^ CD146^+^ and CD146^−^ PCs in 200 mL of EBM + 0.2% BSA were routinely placed on the top of the membrane insert (the upper compartment of the well). To evaluate spontaneous migration, 500 mL of control EBM medium was added to the lower compartment of the wells. To evaluate PDGF-BB-induced CD146^+^ and CD146^−^ PCs migration, 10 ng/mL of PDGF-BB (ReliaTech, Wolfenbüttel, Germany) was added to the lower compartment of each well. A substance placed in the lower compartment of the well acts as a chemoattractant, and the cells move from the surface through the membrane against a concentration gradient. The migration assay was carried out for 8 h at 37 °C in 5% CO_2_. Then, the membrane inserts were removed, fixed in 10% formalin, and stained with Wright’s solution. Cells attached to the upper surface of the filter were removed with a swab, and the cells that migrated across the membrane were counted by microscopically examining the lower surface. Reported data represent the means ± standard deviation (SD) of the number of cells found in each field. At least 5 different fields for each membrane were counted at 10× magnification. 

### 4.10. Cell Proliferation Assay

The specimens of fat tissue (MFAT), freshly obtained from patients, were washed in PBS at 1200 rpm for 5 min. Then, 3 mL of washed tissue was seeded in almost 6 mL of serum-free RPMI-1640 medium in a T25 flask and incubated for 3–4 days at 37 °C in a humidified incubator. At the end of incubation, the CM was collected and an equal volume of fresh serum-free RPMI-1640 medium was added. Separately, 25,000 HUVECs were seeded for overnight incubation in a 24-well plate. The next day, cells were washed with RPMI + 0.2% BSA and treated with the CM in a serum-free RPMI medium. Cells treated with endothelial basal medium (EBM) + 0.5% FBS and diluted two-fold with serum-free RPMI medium were used as a negative control. Cells treated with complete medium (EGM + 10% FBS) were used as a positive control. Following overnight incubation, cells were photographed, trypsinized, and counted using trypan blue exclusion. Images were recorded using an inverted Hitachi KP-D50 camera (Hitachi Ltd., Tokyo, Japan) at 4× magnification.

### 4.11. Tube Formation Assay

The CM from cultured pericytes in EGM-2 complete medium was collected post 3 days of incubation at 37 °C, clarified at 1200 rpm, 5min, and stored at −20 °C. A tube formation assay was used to test the effect of CM from pericytes on vascular morphogenesis of ECs. Briefly, around 50 μL of GFR matrigel (Sigma) was placed into cold wells of a 96-multiwell plate (Corning, NY, USA) at 37 °C for 30 min until jellification. HUVECs were then seeded on Matrigel at a concentration of 10^4^ cells/well in 50 μL of EGM basal medium diluted two times with CM from the corresponding PCs subpopulation. ECs treated with EGM complete medium were used as a control. The number of cords was analyzed, photographed, and counted after 24 h as a parameter of tube formation with an inverted microscope at 10× magnification.

### 4.12. Immunofluorescence Staining for CD56 (NCAM)

Cultures of PCs were seeded on 8-well chamber slides at a confluence of 70% and maintained overnight in culture. For immunofluorescence analysis, cells were fixed with 4% paraformaldehyde in PBS, pH 7.0, permeabilized with 0.5% Triton-X100 in PBS, at RT, and blocked with PBS-BSA5%-NGS2%. Then, cells were stained with a mouse monoclonal antibody anti-CD56 (NCAM/ERIC 1:250; Santa Cruz Biotechnology), and a rabbit polyclonal antibody against PDGFRβ (1:100; Invitrogen), followed by Alexa Fluor-555 donkey anti-mouse and Alexa Fluor-488 goat anti-rabbit secondary antibodies (Thermofisher Scientific). Nuclei were stained with 4′,6-diamidino-2-phenylindole (DAPI) (Thermofisher Scientific). Maximum projection images were acquired via confocal microscopy (C1/TE2000-E microscope; Nikon) using 40× or 100× objectives for evaluation of NCAM and PDGFRβ staining on at least 5 adjacent image fields. Parameters for image acquisition were defined and not modified to allow the comparison of fluorescence intensity as a measure of relative quantification. Image analysis was performed with Image J and FIJI software [42] (NIH, USA).

### 4.13. Statistical Analysis

Results are expressed as mean ± standard deviation (s.d.). Statistical significance was evaluated by one-way analysis of variance following the Bonferroni post-test and Student’s *t*-test. Statistical significance of differences was set at *p*-value < 0.05. Statistical tests were performed using Prism8 software (GraphPad, San Diego, CA, USA).

## 5. Conclusions

MSCs produce and recruit different growth factors to promote tissue regeneration and improve the microenvironment while their ablation or insufficient coverage can lead to abnormal vasculature and leaky cancers. The two immunophenotypically different subpopulations of MSCs were derived from fat tissue also known as pericytes (CD31^–^CD146+) and supra-adventitial adipose stromal cells (SA-ASC; CD31^–^CD146^–^) [29], respectively. Both the populations may tend to develop similar immunophenotypes under similar culture conditions [2,43]. We found that the one expressing the CD146 marker was able to interact better with ECs via higher expression of the NCAM/CD56 adhesion molecule when compared with their CD146 negative counterpart. Consequently, we propose that CD146-expressing pericytes that promote the interaction with endothelial cells can therefore be utilized as a therapeutic target for both repairing unstable vessels as well as newly formed damaged vessels.

## Figures and Tables

**Figure 1 ijms-23-05806-f001:**
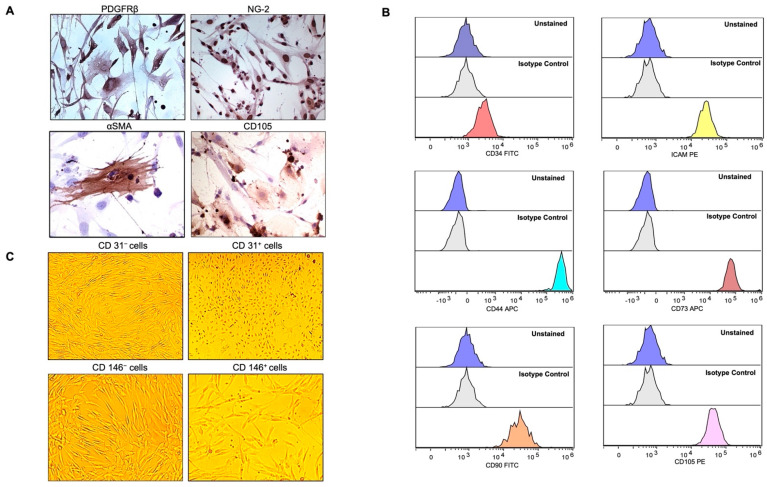
Expression of mesenchymal cell markers (MSCs) on cultured pericytes (PCs). (**A**) Immunohistochemistry staining with CD31^−^ PCs were strongly positive for PDGFRβ, NG-2, and most of the cells were positive for CD105 but slightly for αSMA. Pictures were taken at magnification 40× after avidin-biotin peroxide staining. (**B**) Flow cytometry demonstrated CD31^−^ PCs to be mostly negative for CD34, and positive for ICAM1, CD44, CD73, CD90, and CD105. Histograms in the lowest panels (multicolor) represent the staining for specific antibodies as compared to the unstained and unrelated isotype-matched antibodies in blue and light grey histograms, respectively. (**C**) Bright-field microscopy depicting the morphology of CD31^−^ PCs versus CD31^+^ cells as the latter were distinguished by the endothelial-cell-like cobblestone morphology from the elongated and slender shape of the former. CD31^−^ cells, selected for CD146-based MACS (CD146^−^ and CD146^+^), were similar in shape and morphology. Pictures were taken at a magnification of 10×.

**Figure 2 ijms-23-05806-f002:**
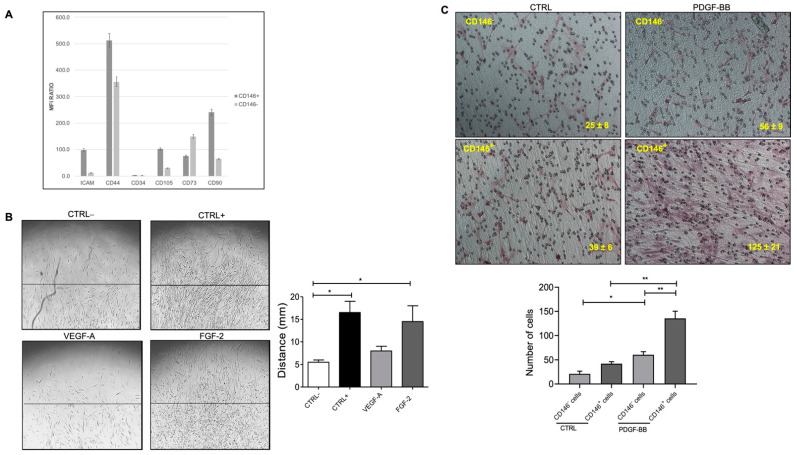
(**A**) The bar graph represents the mean fluorescence intensity (MFI) ratio of each specific antibody and the relative isotype control by flow cytometric analysis of CD146^+^ and CD146^−^ cell populations. Values greater than 1 indicate the expression of the specific marker. (**B**) PDGF-BB induced pericyte migration. The presence of FGF-2 promoted CD31^−^ PCs migration but not VEGF-A as compared to the negative control (EBM-2 + 0.5% FBS). PCs stimulated with complete medium (EGM-2 + 10% FBS) were used as a positive control. The bar chart represents the relative distance migrated outwards from the collagen layer towards the edge of the wells upon stimulation with VEGF-A and FGF-2. At least 3 different fields/well were counted at 10× magnification in triplicates. Scale bar = 100 μM. (**C**) PDGF-BB promoted PCs migration for CD146^+^ cells higher than CD146^−^ cells in the basal medium in a transwell assay. The bar chart represents the number of cells that migrated across the membrane and were counted by microscopically examining the lower surface of the transwell chamber. Reported data represent the means ± standard deviation (SD) of the number of cells found in each field. At least 5 different fields for each membrane were counted at 10× magnification. Scale bar = 100 μM. Images are representative of at least three independent experiments with similar results performed in triplicates. Statistical analysis was performed by one way ANOVA and Bonferroni’s post-test; * *p* < 0.05, ** *p* < 0.01. CTRL^+^—positive control; CTRL^−^—negative control; VEGF-A—vascular endothelial growth factor-A; FGF-2—fibroblast growth factor-2; PDGF-BB—platelet-derived growth factor-BB.

**Figure 3 ijms-23-05806-f003:**
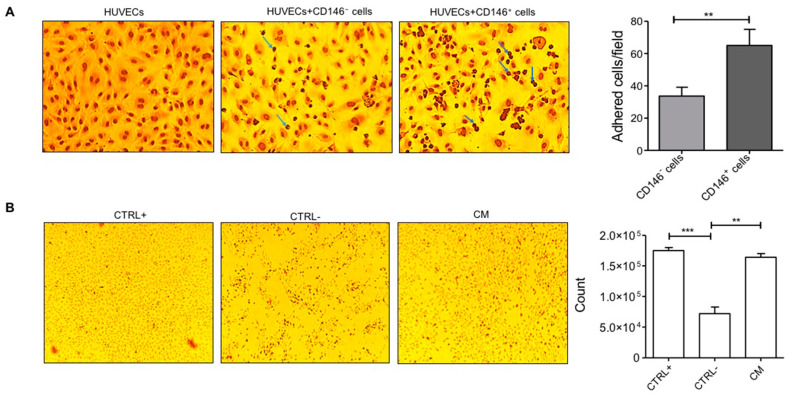
ECs-PCs interaction (**A**) Adhesion assay. PCs detached from tissue culture flasks were seeded on a cultured ECs monolayer. In the figure, adherent PCs appear as darker round cells (for example, indicated by blue arrows) adhered to the elliptical-shaped HUVECs monolayer. CD146^+^ cells were relatively more adherent than CD146^−^ cells to ECs. Images are representative of three different experiments performed in triplicates. The left corner image represents the HUVECs monolayer as a negative control in the absence of adhered pericytes. The bar graph displays the number of adhered cells per frame (counted for three different frames/well) performed in triplicates at 10× magnification. (**B**) Cell proliferation assay. Cells cultured under starvation conditions proliferated in the presence of CM from MFAT (diluted 2-fold in RPMI medium). Images were taken after 16 h of treatment at 4× magnification. ECs cultured in the complete medium (EGM + 10% FBS) were used as a positive control. ECs cultured under starvation conditions (EBM + 0.5% FBS; NC) were used as a negative control. The bar graph shows the cell count after overnight treatment with CM. Images are representative of one out of three independent experiments with similar results performed in triplicates. Statistical analysis was performed by Student’s *t*-test and one way ANOVA following Bonferroni’s post-test; ** *p* < 0.01, *** *p* < 0.001. CM—conditioned medium.

**Figure 4 ijms-23-05806-f004:**
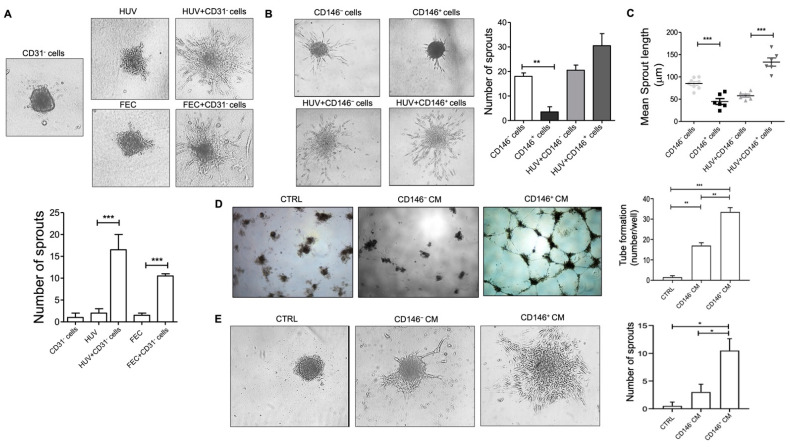
Sprout formation from ECs and PCs co-culture spheroids. (**A**) Representative images of sprout formation from ECs-only (HUVECs or CD31^+^ Fat ECs), and CD31^–^ PCs:ECs (1:5) co-culture spheroids embedded in type I collagen gel at 24 h (magnification, 20×). The quantitative graph shows sprout numbers formed from CD31^−^ PCs- or ECs-only (HUVECs or CD31^+^ fat ECs), compared to co-culture spheroids (mean ± SD, *n* = 3). (**B**) Sprout formation from CD146^−^ and CD146^+^ PCs-only, and PCs + ECs (HUVECs) co-culture spheroids in collagen after 24 h of embedding (magnification, 20×). The quantitative graph displays the number of sprouts at 24 h from PCs-only and PCs:ECs (1:5) co-culture spheroids (mean ± SD, *n* = 3). (**C**) The scatter plot displays the mean sprout length (in µm) per spheroid at 24 h (mean ± SD, *n* ≤ 6). Images are representative of one out of three independent experiments with similar results performed in triplicates. (**D**) HUVECs were seeded on BME-coated plates and stimulated with CM from CD146^−^ and CD146^+^ pericytes in culture. Images were taken after 24 h of plating the assay (original magnification, 10×). HUVECs treated with EGM complete medium were used as a control. The number of cords was counted as a parameter for the quantification of tube formation. (**E**) The 3-D spheroids assay of HUVECs in the presence of CM from CD146^−^ and CD146^+^ cells. HUVECs treated with EGM complete medium were used as a control. Images were taken after 24 h at 10× magnification. Images are representative of one out of two independent experiments with similar results performed in triplicates. Statistical analysis was performed by one way ANOVA and Bonferroni’s post-test; * *p* ≤ 0.05, ** *p* < 0.01, *** *p* < 0.001. HUV—human umbilical vein endothelial cells; FEC—endothelial cells from fat tissue.

**Figure 5 ijms-23-05806-f005:**
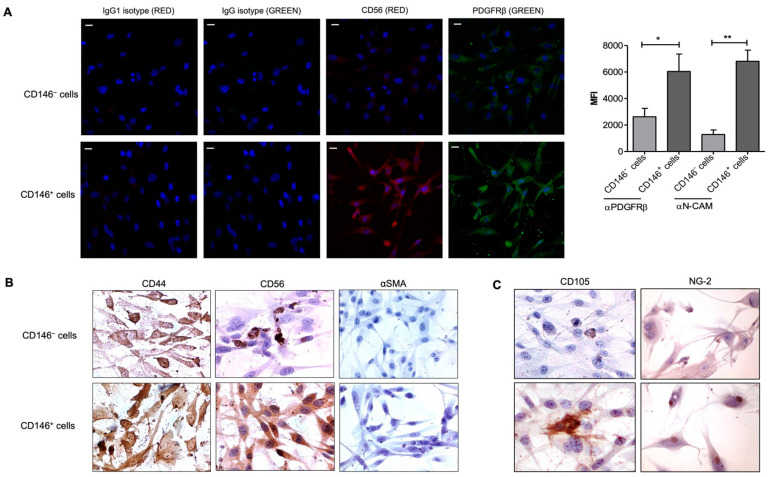
Expression of CD56 (NCAM). (**A**) Representative confocal images of CD146^−^ cells (upper side) and CD146^+^ (lower side). Immunofluorescence was performed to identify PDGFRβ and CD56 expression by both the cell subpopulations. Images display PDGFRβ in green, CD56 in red, and cell nuclei in blue; magnification, 40×. Scale bar = 25 μM. The corresponding IgG- (green) or IgG1-isotype (red) was used as a negative control. The bar graph was generated using the mean fluorescence intensity of two different experiments performed in triplicates. One-way ANOVA and Bonferroni’s post-test were used to compare the data; * *p* ≤ 0.05, ** *p* < 0.01. (**B**) CD146^−^ (upper) and CD146^+^ (lower) cells were stained for peculiar mesenchymal markers CD44, CD56 (NCAM), and αSMA. Images were taken at 60× magnification. (**C**) Immunostaining images represent CD146^−^ (upper) and CD146^+^ cells (lower) for the expression of common mesenchymal markers, CD105 and NG-2 at 40× and 20× magnification, respectively. Images are representative of one out of three experiments performed in duplicates. NCAM—neural cell adhesion molecule (CD56).

**Figure 6 ijms-23-05806-f006:**
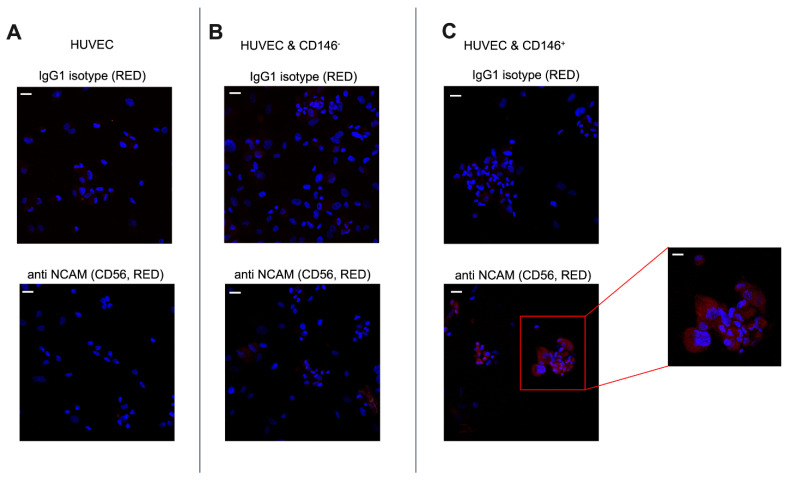
Differential expression of CD56 by CD146^−^ and CD146^+^ pericytes adhered to HUVECs. (**A**) The upper image represents HUVECs stained with an anti-IgG1 isotype. The bottom image represents the HUVECs monolayer stained with antibody to CD56 (NCAM) as a negative control. (**B**,**C**) PCs detached from culture flasks were seeded on the cultured ECs monolayer. The two different morphologies in the figure represent adherent PCs as round cells adjacent to relatively long-shaped HUVECs. The upper images represent ECs-PCs co-culture stained with the negative control IgG1 isotype. The bottom images represent ECs-PCs co-culture stained with CD56 (NCAM). Images are representative of two different experiments performed in triplicates; magnification, 40×. Scale bar = 25 μM. The extreme right image displays the higher magnification image; magnification, 100×. Scale bar = 10 μM. Images display pericytes in red, adhered to HUVECs, and nuclei in blue.

## Data Availability

The data presented in this study are available in this paper and Appendix A.

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
