# Peer review of "CD146^+^ Pericytes Subset Isolated from Human Micro-Fragmented Fat Tissue Display a Strong Interaction with Endothelial Cells: A Potential Cell Target for Therapeutic Angiogenesis"

_ijms, 2022, doi:10.3390/ijms23105806_

Round 1

Reviewer 1 Report

 Your article is significantly improved, thank you for accepting of suggestions. All congratulations to all the efforts and job you performed.

Author Response

We thank the reviewer for his/her constructive comments in improving the manuscript and its further consideration for the purpose of the publication.

Reviewer 2 Report

After some research it appears you can do statistics on n=2, but specific tests are required. I am not a statistician. To make sure the authors are using the correct statistical measures they really need to consult a statistician.

Not withstanding the statistics, I am  satisfied by the authors comments. However, minor changes are required.

  1. Please define acronyms at first instance. There are  acronyms defined in the methods but these come after the results/data e.g., MFI. Please check that you have not missed anything.
  2. How many cells were cultured in proliferation assays? Include in methods
  3. How many pericytes and HUVECs were used to make a 1:5 ratio for the spheroid assays? Include in methods.
  4. Fig 4D and E graphs are misleading. HUVECS are in all treatment conditions, however in the graph, it looks like HUVECs are only in the first bar.

Author Response

We thank the reviewer for his/her constructive comments and further consideration of the manuscript. We have made the following changes as per the suggestions. We hope you will find the modified version of the manuscript suitable for the purpose of publication.

  1. Please define acronyms at first instance. There are  acronyms defined in the methods but these come after the results/data e.g., MFI. Please check that you have not missed anything.

  • Thank you for the comment. All the acronyms have been expanded at the first place only in the main text of the manuscript. All changes have been highlighted.

  1. How many cells were cultured in proliferation assays? Include in methods

  • The number of cells used in proliferation assay is mentioned in the methods section (Page-13, Line-510). We have highlighted the information.

  1. How many pericytes and HUVECs were used to make a 1:5 ratio for the spheroid assays? Include in methods.

  • Thank you for the suggestion. We have now added and highlighted the imperative information in the methods section (Page-12, Line-451).

  1. Fig 4D and E graphs are misleading. HUVECS are in all treatment conditions, however in the graph, it looks like HUVECs are only in the first bar.

  • We agree with the reviewer’s comment. We have changed the captions in the images and bar graph of Fig. 4D and 4E by replacing the term “HUVECs” with “Control”.